# Effects of Rumen-Protected L-Tryptophan Supplementation on Productivity, Physiological Indicators, Blood Profiles, and Heat Shock Protein Gene Expression in Lactating Holstein Cows under Heat Stress Conditions

**DOI:** 10.3390/ijms25021217

**Published:** 2024-01-19

**Authors:** Jang-Hoon Jo, Ghassemi Nejad Jalil, Won-Seob Kim, Jun-Ok Moon, Sung-Dae Lee, Chan-Ho Kwon, Hong-Gu Lee

**Affiliations:** 1Department of Animal Science and Technology, Sanghuh College of Life Sciences, Konkuk University, Seoul 05029, Republic of Korea; godandthegod@naver.com (J.-H.J.); jalilgh@konkuk.ac.kr (G.N.J.); 2Department of Animal Science, Michigan State University, East Lansing, MI 48824, USA; kws9285@hanmail.net; 3Institute of Integrated Technology, CJ CheilJedang, Suwon 16495, Republic of Korea; junok.moon@cj.net; 4Animal Nutrition and Physiology Team, National Institute of Animal Science, Rural Development Administration, Wanju 55365, Republic of Korea; leesd@korea.kr; 5Department of Animal Science, Kyungpook National University, Sangju 37224, Republic of Korea; chkwon@knu.ac.kr

**Keywords:** heat stress, rumen protected L-tryptophan, lactating Holstein cow, welfare

## Abstract

In this study, we examined the effects of rumen-protected L-tryptophan supplementation on the productivity and physiological metabolic indicators in lactating Holstein cows under heat stress conditions. The study involved eight early lactating Holstein cows (days in milk = 40 ± 9 days; milk yield 30 ± 1.5 kg/day; parity 1.09 ± 0.05, *p* < 0.05), four cows per experiment, with environmentally controlled chambers. In each experiment, two distinct heat stress conditions were created: a low-temperature and low-humidity (LTLH) condition at 25 °C with 35–50% humidity and a high-temperature and high-humidity (HTHH) condition at 31 °C with 80–95% humidity. During the adaptation phase, the cows were subjected to LTLH and HTHH conditions for 3 days. This was followed by a 4-day heat stress phase and then by a 7-day phase of heat stress, which were complemented by supplementation with rumen-protected L-tryptophan (ACT). The findings revealed that supplementation with ACT increased dry matter intake as well as milk yield and protein and decreased water intake, heart rate, and rectal temperature in the HTHH group (*p* < 0.05). For plateletcrit (PCT, *p* = 0.0600), the eosinophil percentage (EOS, *p* = 0.0880) showed a tendency to be lower, while the monocyte (MONO) and large unstained cells (LUC) amounts were increased in both groups (*p* < 0.05). Albumin and glucose levels were lower in the HTHH group (*p* < 0.05). The gene expressions of heat shock proteins 70 and 90 in the peripheral blood mononuclear cells were higher in the ACT group (HTHH, *p* < 0.05). These results suggest that ACT supplementation improved productivity, physiological indicators, blood characteristics, and gene expression in the peripheral blood mononuclear cells of early lactating Holstein cows under heat-stress conditions. In particular, ACT supplementation objectively relieved stress in these animals, suggesting that L-tryptophan has potential as a viable solution for combating heat-stress-induced effects on the cattle in dairy farming.

## 1. Introduction

Heat exposure in ruminants decreases their productivity and physiology, depressing milk production [1]. The well-acknowledged factor through which heat stress (HS) affects modern dairy cows is reduced dry matter intake (DMI), which leads to reduced nutrient absorption and increased basal metabolism (maintenance requirements) caused by activation of the thermoregulatory system [2].

Despite implementing heat-abatement strategies, animal productivity remains suboptimal during the warm summer months. Heat stress (HS) jeopardizes animal health, which is mediated in part by reduced intestinal barrier integrity, which is known as the leaky gut [3]. During HS, blood is directed to the periphery and skin to maximize heat dissipation. Thus, implementing complementary management and dietary approaches is necessary to allow animals the opportunity to maximize their productivity during HS. Feeding diets with balanced amino acid proportions in metabolizable protein can increase lactation performance and milk protein and fat concentration while enhancing the response to stressful conditions under decreased DMI [4].

In particular, identifying the dietary molecules that protect gut barrier health could profoundly enhance animal plasticity. Supplementation with rumen-protected L-tryptophan (ACT) improves the intestinal barrier function in a variety of ruminants [5,6]. Additionally, ACT in the form of an acetate and an L-tryptophan complex mitigates the hyperpermeability effects of HS and feed restriction on inflammation and intestinal health in Hanwoo steer [6]. This may contribute to the decomposition of ACT into acetate and L-tryptophan in the liver, potentially increasing energy and protein expenditure efficiency. Moreover, Choi et al. [7] observed an increase in milk parameters, including yield and protein content, when dairy cows subjected to HS in an external environment were fed ACT. Beyond generating serotonin, melatonin, and other bioactive substances, L-tryptophan and its metabolites, including melatonin, serotonin, and kynurenine, play roles in biological functions through antioxidant activity and stimulating heat shock protein (HSP) expression to counteract the damage caused by free radicals and protein denaturation [8]. Within this context, we supplemented the feed of early lactating Holstein cows with L-tryptophan to assess its direct benefits on the health status and productive performance of these cows. However, few studies have been conducted on a supplementation of ACT under HS conditions related to productivity and physiological indicators. In this study, we meticulously examined the physiological metabolic changes in dairy cows within metabolic chambers, aiming to increase the precision of previous findings about the optimal concentration of ACT supplementation in external environments and assessing the impact of ACT on parameters like DMI, milk yield and composition, blood hematology and metabolites, and HSP gene expression under HS conditions, to evaluate the overall effectiveness of ACT in dairy cow diets.

## 2. Results

### 2.1. Feed and Water Intake and Milk Yield

The DMI in early lactating Holstein cows was decreased by 2% in the LTLH treatment and by 30% in the HTHH treatment but was increased with the addition of ACT by 1% and 20% in the LTLH and HTHH treatments, respectively (*p* < 0.05). Moreover, the WI was decreased by approximately 0.3 kg/day when ACT was fed at LTLH, but it was decreased by about 30 kg/day at HTHH (*p* < 0.05). The DMI and WI ratio was decreased in HS conditions when ACT was fed. Specifically, the change was significant in the HTHH treatment (*p* < 0.05, Table 1). In addition, under LTLH and HTHH conditions, milk yield was increased by 0.5 and 0.3 kg/day, respectively, when ACT was fed (*p* < 0.05, Table 1). A significant interaction was found between HS and ACT for the DMI, WI, and feed and water ratio (*p* < 0.05), and a tendency toward significance was found for milk yield (*p* = 0.0880).

### 2.2. Milk Characteristics

Although no significant difference in milk protein was found among the cows, the milk protein content was decreased under HS in the HTHH group and showed a tendency to increase when ACT was fed (*p* = 0.0523, Table 2). However, HTHH treatment resulted in a decrease in 3.5% FCM and ECM. A significant interaction was found between HS and ACT for the milk urea nitrogen (MUN; *p* < 0.05) content.

### 2.3. Blood Hematology

The supplementation with ACT resulted in a significant increase in the numbers of monocytes (MONOs) and large unstained cells (LUCs) in both groups (*p* < 0.05), as determined from the results of blood hematology (Table 3). A decreasing trend was observed in plateletcrit (PCT; *p* = 0.0600) and eosinophil percentage (EOS, *p* = 0.0880), whereas the LUC (*p* = 0.0656) showed an increasing trend (Table 3). Conversely, HTHH treatment led to a significant decrease in red blood cell (RBC) count and mean corpuscular volume (MCV; *p* < 0.05, Table 3). Moreover, a decreasing trend was observed in hematocrit (HCT; *p* = 0.0785) content, whereas the mean corpuscular hemoglobin (MCH; *p* = 0.0696) and basophil (BASO) content (*p* = 0.0594) showed an increasing trend (Table 3).

### 2.4. Blood Metabolites and Hormones

Albumin and glucose concentrations were increased and the ghrelin concentration was decreased under HS conditions (LTLH and HTHH, *p* < 0.05, Table 4). However, albumin and glucose concentrations were decreased when L-tryptophan-containing ACT was fed to the cows. The cortisol concentration increased with increasing HS severity (HTHH) and decreased when the feed was supplemented with ACT. Although no statistically significant difference was found in the haptoglobin levels in relation to the inflammatory response, the levels increased under HS conditions and tended to decrease when ACT was added to the feed (*p* = 0.0765). A significant interaction was identified between HS and ACT for the albumin and ghrelin concentrations (*p* < 0.05), and there was a tendency with blood urea nitrogen (BUN; *p* = 0.0521) and cortisol (*p* = 0.0687) concentrations.

### 2.5. Gene Expression

#### 2.5.1. Peripheral Blood Mononuclear Cell (PBMC)

The supplementation with ACT in the LTLH and HTHH groups resulted in a significant upregulation of HSP70 expression (*p* < 0.05, Table 5). Although no significant statistical difference was found between the groups, we observed that HSP90 expression was numerically higher in the HTHH group. A significant interaction was found between HS and ACT for HSP70 expression (*p* < 0.05).

#### 2.5.2. Hair Follicle

The supplementation with ACT in both LTLH and HTHH groups did not result in any statistically significant differences in the expression levels of HSP90 (*p* > 0.10), as shown in Table 5.

### 2.6. Physiological Indicators

The HR and RT, which are widely recognized as indicators of HS, significantly increased (*p* < 0.05, Table 6), indicating that the animals were experiencing HS. The supplementation with ACT led to a decrease in the HR and RT values in both the LTLH and HTHH groups with the HTHH group showing greater decreases (−2% vs. −5%) compared with the LTLH group (−1% vs. −2%).

## 3. Discussion

### 3.1. Feed and Water Intake and Milk Yield

The relationship between increasing environmental temperature, body temperature, and reduced DMI has been well established [9]. However, limited research has been conducted on how reductions in DMI or changes in feed composition during HS affect the productivity of cows. Decreased appetite and reduced nutrient availability during HS can cause substantial weight loss, leading to a physiologically negative energy balance [10] and a decline in body condition score. Environmental stress caused by factors such as intestinal barrier problems and inflammation can lead to intestinal infections and impaired performance [11].

Ominski et al. [12] reported a decrease in DMI after 4–5 days of heat exposure. The 2 days prior to heat exposure were found to most strongly affect DMI [13], which agrees with the findings reported by Collier et al. [14]. A simulated heat wave (29 °C, ~50% relative humidity for 4 days) suppressed DMI as early as 1 day after the rise in temperature [15]. This decrease in DMI is likely an inherent response by cows to reduce the heat produced via feed metabolism, which thereby affects milk production [16]. In this study, we found that the L-tryptophan in ACT alleviated the effects of HS and increased DMI by 6 kg/day. ACT supplementation can increase DMI in dry cows [5], livestock, and poultry [17], and it can also improve growth and gut integrity [18]. Additionally, the feeding of L-tryptophan enhances intestinal cell protein turnover, tight junction protein expression [19], and microbiota diversity [20], thereby improving intestinal barrier function. An alternative explanation is that the incorporation of L-tryptophan with acetate may have reduced rumen digestion and enhanced small intestinal digestion and absorption [6,21]. Ensuring adequate levels of L-tryptophan in the diet can increase serotonin levels in the brain and improve DMI by animals [17].

The hormone ghrelin, which is produced by the cells in the gastrointestinal tract, is released when the stomach is empty to promote hunger and stimulate gastrointestinal motility [22]. Although information concerning the direct effects of HS on ghrelin synthesis is limited, existing reports suggest that acylated and total ghrelin levels are generally suppressed during summer months in both primiparous and multiparous Holstein cows [23]. Intensified cooling management led to increased DMI, improved welfare parameters, and higher concentrations of total and acylated ghrelin [24]. Acylated ghrelin is a potent orexigenic signal in various species, including ruminants [25]. Therefore, the reduced feed consumption during the summer months could be linked to lower levels of ghrelin (total and/or acylated), which could dampen the animals’ DMI [23].

Serotonin can suppress ghrelin secretion, which is potentially influenced by its precursor L-tryptophan. Brain serotonin systems, especially through serotonin receptor 2C, regulate eating behavior and energy homeostasis, with the serotonin receptor 2C and serotonin receptor 1B mediating anorexic actions of m-chlorophenylpiperazine and D-fenfluramine. This suggests a negative feedback mechanism between brain serotonin and plasma ghrelin levels [26]. Dorsal raphe nucleus serotonin neurons are linked to feeding suppression, which is modulated by metabolic hormones [27]. The exact relationship between L-tryptophan concentration and ghrelin levels remains unclear due to factors such as dosage, administration, and the levels of other nutrients. The endocrine system’s intricate interactions pose challenges for understanding these relationships. Even when DMI increases, the role of ghrelin appears to remain stable; however, examining daily variations could yield additional insights. Supplementation of ACT to dairy cows subjected to HS induces a multifaceted response, which is characterized by the involvement of the ghrelin hormone. This response plays a critical role in reducing internal heat production and enhancing DMI, which typically diminishes in the presence of stress.

Heat stress (HS) increases WI in dairy cows [1,28] and Korean native beef steers [29] to compensate for sweat-induced water loss. Water losses through respiratory tract activity and thermoregulation mechanisms inherently result in an increased WI, which subsequently may lead to a decrease in DMI. L-tryptophan accelerates hydrate formation, reaching 90% conversion in 30 min [30]. In this study, an unexpected decrease in WI was observed with ACT supplementation, although the underlying mechanism for this finding remains unclear. L-tryptophan can induce cholecystokinin (CCK) secretion [31], which activates CCK-positive neurons in the subfornical organ (SFO) via the CCK-B receptor. This activation results in the inhibition of WI through gamma-aminobutyric acid (GABA)ergic interneurons. Cues from WI modulate SFO neurons, which are central to thirst regulation. Importantly, the modulation of CCK-positive SFO neurons can reduce WI even in a state of water satiety. Water ingestion transiently activates these neurons, leading to the suppression of water-sensitive neurons in the SFO, potentially connecting the reduction in WI to energy satiety. A similar decrease in WI was noted with L-tryptophan supplementation in pigs [32] and broilers [33]. These dose-dependent effects on energy intake also influence thirst behavior. Fos expression, indicating the swift inhibition of CCK-positive neurons during periods of water depletion, underscores L-tryptophan’s potential role in modulating both thirst and feeding. The mechanisms of CCK neuron activation are unclear. Comprehensive research is needed to understand body fluid homeostasis, focusing on intricate neural systems and their integration for optimal behavioral outcomes.

The quantity and quality of absorbed amino acids are crucial for milk production In dairy cows under HS conditions [7,34]. The availability of L-tryptophan stimulates the synthesis of brain serotonin and melatonin, which is associated with prolactin release [35]. Miao et al. [17] demonstrated that feeding 0.12% L-tryptophan to pigs increased milk yield. Ma et al. [34] reported that supplementing the diets of 24 dairy cows with 14 and 28 g/day of L-tryptophan led to increased milk yield. In addition, Choi et al. [7] found that milk production increased compared with that of the control group when cows were supplemented with 30 g/day of ACT during summer, which is a period characterized by HS. Consequently, it can be deduced that supplementation with ACT is efficacious in counteracting the decrease in milk yield observed in conditions of HS.

The decrease in DMI was previously reported to be the primary factor leading to reduced milk yield during HS [36]. However, other physiological and metabolic factors are known to contribute to this phenomenon [37]. The 95% absorption of ACT into the small intestine via ruminal bypass may have led to its decomposition into acetate and L-tryptophan in the liver, potentially contributing to higher-efficiency energy and protein expenditure [6]. Additionally, the acetate derived from ACT may serve as a source of energy for milk fat synthesis and the TCA cycle, leading to an increase in milk fat concentration, as suggested by Urrutia et al. [38].

Therefore, including rumen-protected L-tryptophan (ACT) in the diet not only reduces the supply of nondegraded protein feeds and enhances the efficiency of amino acid use but also increases the total milk yield of lactating cows under HS conditions. Compared to previous studies, we treated HS under constant THI conditions through a metabolic chamber and collected more precise data by supplementing ACT. Consequently, there is a pronounced need for further research to elucidate the mechanisms through which ACT mediates a decrease in WI and an elevation in DMI via the ghrelin hormone pathway and to explore the resultant implications on enhanced milk yield.

### 3.2. Milk Characteristics

Milk protein concentration decreases in cows subjected to high THI environments [16]. These changes in milk composition are not due solely to a reduction in DMI but also to a reduced supply of protein precursors to the mammary gland and an increase in the use of amino acids for other biochemical processes, such as gluconeogenesis and the synthesis of acute phase and HSPs [39].

The quantity of protein in milk is influenced by the amino acid balance, composition, and intake in the diet, which can be enhanced by adding crude protein and rumen-undegradable protein [40]. However, under HS conditions, dairy cows exhibit reduced DMI and suppressed rumination activity as a means to decrease the heat produced by the body. Insufficient L-tryptophan in the diet also reduces muscle and liver protein synthesis in weaned piglets [41]. Heat shock can disrupt lactation homeostasis by affecting serotonin signaling in mammary gland receptors [42]. Providing adequate L-tryptophan is crucial for optimal animal growth and intestinal homeostasis, activating mammalian target of rapamycin and enhancing tight junction protein expression in porcine intestinal epithelial cells [43]. L-tryptophan supplementation improved the production of milk proteins and the metabolic processes for energy within bovine mammary cells [44]. In addition, supplementing dairy cows with rumen-protected L-tryptophan during HS enhanced milk protein concentrations [7]. In our study, supplementation with ACT under HS conditions demonstrated a statistically significant tendency to increase milk protein levels. To clarify the results, a larger-scale experiment could provide more detailed insights.

The transcriptional level of beta-casein expression in HC cells increased with mild heat treatment (39 °C) without inducing differentiation. Compared with cells cultured at 37 °C, those cultured at 39 °C showed significantly higher expression levels of X-box binding protein 1 and activating transcription factor 6 alpha [45]. Furthermore, the knockdown of XBP1 or ATF6α using siRNA in HC cells resulted in the suppression of β-casein mRNA expression levels induced by mild heat treatment [45]. These findings demonstrate the involvement of ATF6a and XBP1 in the upregulation of beta-casein expression upon mild heat treatment.

Feed efficiency evaluation in dairy herds based solely on milk volume per unit of feed consumed may be misleading, particularly in cows experiencing HS, due to metabolic changes that can alter mammary gland metabolism and decrease milk component synthesis. Therefore, feed efficiency should be evaluated based on ECM and FCM. Knapp et al. [46] reported a decrease in FCM of 3.5% when dairy cows were exposed to 31 °C at 60% humidity. Choi et al. [7] reported that supplementing early lactating Holstein cows with 30 g/day rumen-protected L-tryptophan increased FCM and ECM. Our results also showed that 30 g/day ACT supplementation led to a higher milk fat concentration, 3.5% FCM, and ECM compared with those in the LTLH group, which may have partly been due to the increased DMI produced by ACT feeding. Balanced amino acid feeding may increase fatty acid synthesis in the mammary glands by increasing the mRNA expression of lipogenic enzymes, thereby increasing milk fat yield [47]. However, the severe stress level in the HTHH group may have prevented any increase in FCM or ECM despite the supplementation with ACT. Further studies are needed to clarify the underlying metabolic processes involved in the above-mentioned phenomena.

Our research findings have shown that the supplementation of ACT during HS improves milk protein. Based on these results, it is thought that high-quality milk and products can be produced even under HS conditions. It is determined that further research is needed on the mechanism involving acetate and L-tryptophan in ACT, which are related to milk yield and milk protein synthesis, including beta-casein and the mammalian target of rapamycin pathway, during HS situation.

### 3.3. Blood Hematology

The results of hematological analysis aid with diagnosing and monitoring various diseases, with hematology focusing on the physiology, pathology, and prognosis of blood-related disorders.

Elevated environmental temperatures can lead to reduced RBC counts in dairy cattle due to hemodilution owing to increased water intake for cooling [48]. Similarly, Ondruska et al. [49] observed reduced RBC counts in rabbits exposed to 36 °C for 12 h daily spans over 4 weeks. HS-induced RBC count reductions could stem from thyroid and lymphatic system dysfunction or cell membrane damage due to free radicals or nutrient deficiencies [50]. Oxidative stress from increased free radicals can harm lipid-rich organelles and MCV, impairing cellular functions [51]. Consequently, hemoglobin’s oxygen-binding ability diminishes during HS, affecting tissue health beyond the critical thermal maximum [52]. Azeez et al. [53] noted that high-temperature and high-humidity exposure caused a lower MCV in dogs with MCV reflecting RBC count and oxygen content. A reduced MCV could result from enhanced RBC destruction or hindered erythrocyte production [54], often signifying microcytic anemia.

The innate immune system comprises cells such as monocytes, macrophages, and neutrophils that promptly target and eradicate pathogens at infection sites. Monocytes, the primary type of leukocytes, differentiate into macrophages or dendritic cells based on environmental cues and activating signals. L-tryptophan breakdown restricts this amino acid, suppressing T-cell proliferation. L-tryptophan catabolism produces metabolites thought to inhibit immune cells, possibly via proapoptotic pathways [55]. T-cell progression is halted by L-tryptophan scarcity in the G1 phase; L-tryptophan depletion in tissues, induced by indoleamine 2,3-dioxygenase, predisposes T cells to enter apoptosis [56]. Inflammatory cells express serotonin receptors, with serotonin influencing monocyte differentiation into dendritic cells, impacting the expression levels of the associated molecules. Moreover, serotonin regulates chemokine release in human-monocyte-derived dendritic cells: mature dendritic cells exhibit a decreased production of interferon gamma-induced protein 10/C-X-C motif chemokine ligand 10, which is a potent chemoattractant for type 1 T helper cells, whereas immature and mature dendritic cells display increased secretion of C-C motif chemokine ligand 22/macrophage-derived chemokine. Thus, ACT supplementation counteracts the adverse effects of HS by enhancing immune activation.

PCT is used to evaluate the proportions of platelets in the blood. L-tryptophan supplementation may bolster cardiovascular function by lowering blood pressure and enhancing vascular endothelial adaptability to variations in blood flow. Despite ACT supplementation producing no significant differences, increases in RBC, LUC, and monocyte counts and decreases in PCT, EOS, and MCV were observed. The results of hematological assessments revealed no marked changes, suggesting that ACT inclusion beneficially influenced productivity and alleviated the effects of HS without negatively impacting lactating Holstein cows.

### 3.4. Blood Metabolites and Hormone

Evaluating the levels of blood metabolites linked to energy, protein, and mineral metabolism provides insights into dairy animal health and can increase their productivity [1,57]. Metabolic assessment is vital for herd management, assessing nutritional status, monitoring health, detecting subclinical diseases, and addressing metabolic-disorder-related issues [57].

Although multiple conditions, including nephrotic syndrome and hepatic cirrhosis, induce hypoalbuminemia, the leading cause remains the inflammatory response. Elevated rearing temperatures augment plasma glucose and albumin levels. Despite stable albumin levels in HS broilers at 34 °C, goats exposed to prolonged HS (over 15 days) had elevated albumin levels [58]. Similarly, Helal et al. [59] observed that under high THI conditions, the albumin concentrations in Balady and Damascus goats increased by 7.11% and 6.62%, respectively. Albumin, in addition to signifying nutritional status in ruminants [60], primarily aids in ligand transportation and acts as a chaperone to prevent protein misfolding or the aggregation of other proteins. During HS, albumin–fibronectin interaction becomes essential for protein fold correction, which is a mechanism further validated by the therapeutic role of albumin and HSP70 in protein repair [1].

Stress and fatigue elevate catecholamine levels, inhibiting insulin secretion and enhancing lipolysis, thereby raising the levels of free fatty acids in the plasma and altering tryptophan’s binding affinity to albumin [61]. This change may elevate serum 5-hydroxy-tryptamine and serotonin levels, even when the albumin concentration is consistent. Reduced plasma albumin levels can also increase free tryptophan levels [62]. Serum albumin’s conformational changes may regulate brain tryptophan levels, correlating with plasma serotonin alterations [63]. Additionally, increasing WI and HSP induction might decrease stress and inflammation but can lower serum albumin levels.

Rising environmental temperatures increase kinetic energy, enhancing enzyme–substrate interactions and glucose production. L-tryptophan, an essential amino acid, regulates inflammation and insulin resistance through its metabolites [64]. Its complex effects on blood glucose involve metabolites such as serotonin, which modulate insulin secretion and muscle glucose uptake [65]. L-tryptophan dampens post-meal glucose spikes, aiding in beta-cell function [66]. Our data showed that higher ACT levels significantly decreased blood glucose levels. Serotonin stimulates the serotonin receptor in rat adrenal glands, enhancing β-endorphin release and glucose uptake in the muscles [67]. Additionally, serotonin increases glucose uptake in the muscles and liver through cyclin-dependent-kinase-5 activation, promoting glycogen synthesis [68]. L-tryptophan, a nicotinamide adenine dinucleotide (NAD+) liver precursor [69], is pivotal for glucose metabolism. Both the nicotinamide adenine dinucleotide hydrogen (NADH)/NAD+ ratio and the amount of NADH are crucial in this regard with NAD+ synthesis from L-tryptophan potentially enhancing glycogen synthesis [68]. Bruschetta et al. [70] noted a strong negative correlation (r = −0.77, *p* < 0.05) between plasma tryptophan and glucose levels, linking stress to lactation energy deficits. ACT supplementation may enhance glucose absorption in the organs, counteracting the effects of stress, as ACT, after conversion into serotonin and NADH, may serve as an energy source.

In cows exposed to HS, elevated cortisol concentrations signal lactation-related stress, which accelerates lipolysis and possibly increases NEFA concentrations [1]. This stress response may inhibit genes related to T-cell activation and cytokine production, potentially affecting the cellular immune system [71]. Cortisol regulates the immune system by targeting genes associated with cytokines, chemokines, and inflammatory proteins and receptors [1]. Additionally, the increased cortisol levels in the summer may contribute to elevated monocyte counts [72]. Supplementation with ACT reduced cortisol levels [7], possibly through a stress-relieving mechanism similar to that of increased serotonin levels [73]. The kynurenine/tryptophan ratio positively correlated with cortisol in cows with inflammation [74], whereas a stressed condition, associated with lactation-induced energetic deficits, was indicated by a negative correlation between plasma tryptophan and cortisol (r = −0.83, *p* < 0.05; [70]).

Acute-phase proteins from the liver serve as valuable biomarkers of chronic inflammation, which is indicated by decreased negative acute-phase proteins and increased positive-phase proteins such as haptoglobin [75]. We found that HS cows had higher serum haptoglobin levels than adaptation cows, which is in line with the findings reported by Hamzaoui et al. [76] in dairy goats under HS. Additionally, Le Floc’h et al. [77] noted reduced plasma haptoglobin in pigs on an L-tryptophan diet, suggesting L-tryptophan may moderate inflammation. L-tryptophan metabolism has anti-inflammatory and immunosuppressive effects, aiding with controlling hyperinflammation and inducing long-term immune tolerance [78]. Haptoglobin levels did not statistically differ between our groups, so larger-scale trials are needed to further investigate the correlation of ACT with the functioning of the immune system and its role in reducing the levels of cortisol and haptoglobin during HS.

Supplementation with ACT in dairy cow diets may manage stress and inflammation, as indicated by the regulation of blood glucose and cortisol levels, while acute-phase proteins like haptoglobin, along with the immunomodulatory effects of L-tryptophan, provide insights into chronic inflammation and immune tolerance. Additionally, understanding the levels of blood metabolites related to energy, protein, and mineral metabolism under heat stress can inform strategies to improve the health and productivity of dairy cows.

### 3.5. Evaluation of Heat Shock Protein mRNA Expression

Dairy cattle activate cellular mechanisms to protect against the effects of HS [79], such as the release of HSP by heat shock factor 1 [80], which repair misfolded proteins and protect cells during stress [81,82]. HSP70 is a major mechanism in dairy cattle to prevent apoptosis and protect against cell damage during HS [81]. In strong responders to HS, the PBMCs from immune-phenotype cattle showed an increased expression of HSP70 and cell proliferation, indicating an increase in cellular-protection-related genes [83]. HSPs also regulate molecular processes related to extracellular matrix degradation and assembly [84]. Hyperthermic conditions damage mammary epithelial cell morphology and integrity [85].

The gene expression pattern observed in response to acute heat shock exposure was consistent with that found in prior studies [86]. There is limited literature on the role of serotonin in HS responses, but in our recent study on Holstein cows, we found a higher expression of the HSP70 gene under chronic HS [7]. The metabolites of L-tryptophan, nicotinic acid, and nicotinamide may cause mitochondrial stress, leading to the increased expression of HSP through unfolded protein responses and endoplasmic reticulum stress [87]. Our results suggest that ACT supplementation increases the expression level of HSP under HS conditions, which helps prevent protein denaturation and alleviate stress. The specificity of the findings to dairy cows and the limited understanding of certain mechanisms, such as the role of ACT in HS responses, are notable. Therefore, it is necessary to investigate the mechanism of how acetate and L-tryptophan in ACT affect protein folding in relation to HS. This research can guide the development of targeted breeding and dietary supplement strategies in dairy cows to enhance their resilience against HS, potentially improving their overall health and productivity.

### 3.6. Physiological Indicators

The coping mechanisms and metabolic changes of animals due to HS are influenced by various factors, including behavioral and physiological adjustments that are made to regulate body temperature [88]. Heat stress (HS) can cause an increase in heat load after meals, leading to reduced DMI and negatively affecting milk yield and composition [37]. The primary acute heat loss mechanisms include perspiration and elevated HR, which promote heat dissipation through increased blood flow and vasodilation at the body’s peripheries [89]. RT is commonly used as a physiological indicator of HS and body temperature [1]. Al-Qaisi et al. [90] reported that cows experiencing HS showed a 1.2 °C increase in RT and a 29 breaths/min increase in HR compared with non-heat-stressed cows.

The findings of our study indicate that the effects of acute HS in early lactating Holstein cows can be ameliorated via ACT supplementation, as evidenced by substantial decreases in HR and RT. This finding implies that HR and RT are key indicators of the effects of ACT on cows under HS, and so ACT may be involved in activating heat loss metabolism processes during HS exposure. Notably, few researchers have examined the effects of ACT supplementation on these parameters in HS conditions. Heat stress (HS) can cause decreased FI and increased body temperature in lactating Holstein cows [1]. Consistent with the findings of prior research, our findings demonstrate that L-tryptophan can considerably decrease the RT in Holstein steers when supplemented at 38.5 mg/kg/2 h, as reported by Madoka [91]. Similarly, in rats, the intraperitoneal injection of 200–600 mg/kg L-tryptophan resulted in a decrease in body temperature. Additionally, feeding pigs excess dietary L-tryptophan for 5 days prior to slaughter reduced stress, possibly by increasing hypothalamic neurotransmitter concentrations [92]. Based on the aforementioned results, L-tryptophan is a stress-alleviating substance that is capable of reducing body heat dissipation, a representative physiological indicators of stress, in dairy cows exposed to HS.

## 4. Materials and Methods

### 4.1. Experimental Design, and Animals

The experimental procedure received authorization from the Institutional Animal Care and Use Committee at Konkuk University (Approval No: KU19125). Among the eighty multiparous early lactating Holstein cows, eight cows were selected with closely matched conditions: milk production (30 ± 1.5 kg/day; *p* > 0.05), days in milk (40 ± 9 days; *p* > 0.05), and parity (1.09 ± 0.05, *p* > 0.05). Eight early lactating Holstein cows were placed in experimental chambers (3 m × 4 m × 5 m), with four cows per period suitable for each temperature and humidity condition. For each period, the chamber environment was altered by adjusting the temperature and humidity over a three-day adaptation phase. Following the adaptation phase, the temperature and humidity levels were regulated to create HS conditions for a period of 4 days. After 4 days of HS, we supplemented the feed with ACT during HS treatment for 7 days (Figure 1). Experimental diets were fed two times per day at 0800 h and 1400 h. All cows were fed a common basal diet during the entire experimental period according to NRC nutritional requirements [93]. Table 7 summarizes the amino acid compositions of the basal diet. Fresh water was given five times daily during the experimental period. ACT was administered as a powdered supplement, top-dressed onto the feed, which was then mixed. We used ACT derived from N-acetyl-L-tryptophan (CJ CheilJedang, Seoul, South Korea), which is a form of L-tryptophan synthesized by chemically altering the α-amino group with an acetyl group: 95% of this ACT is absorbed into the small intestine through ruminal bypass. Our results demonstrate the strong protective effect of the N-acetyl group in ACT, which effectively prevents the bacterial degradation of L-tryptophan. The results of batch culture experiments using artificial ruminal fluid revealed an escape ratio of approximately 95%. A dose of 30 g ACT per animal facilitated appropriate amino acid balancing in the diet, as substantiated by the pertinent literature [7].

### 4.2. Temperature–Humidity Index

Surrounded by three contiguous seas, South Korea experiences notably hot and humid summer conditions. An extensive 20-year meteorological analysis from 1998 to 2018 demonstrated that South Korea has an average summer temperature of 25.29 °C with a mean minimum of 22.09 °C and a mean maximum of 29.41 °C; the average relative humidity was 78.10% during the summer months (Korea Meteorological Administration, 2018). In a 2020 meteorological survey evaluating temperature and humidity variances across distinct regions in Korea, the findings demonstrated that the prevailing average summer temperature fluctuated between 22 and 25 °C. Simultaneously, relative humidity (RH) exhibited variations, ranging from 70% to 95% [94].

The calculation of the temperature–humidity index (THI) was conducted using the formula (1.8 × T_db_ + 32) − [(0.55 − 0.0055 × RH) × (1.8 × T_db_ − 26)] [95], where T_db_ represents the dry bulb temperature (°C), and RH denotes the relative humidity (%). The experiment was conducted in an environment where temperature and humidity were an automatic controlled system in the experimental chamber using a computer. The chambers were automatically maintained at a fixed ambient temperature of 25 °C (low temperature, LT) and 31 °C (high temperature, HT). Relative humidity (RH) levels were categorized into two ranges: 35% to 50% (low humidity, LH) and 80% to 95% (high humidity, HH) [1]. Temperature regulation within the experimental environment was achieved through the use of an air conditioner (CSVR-Q118E, Carrier Corporation, Seoul, South Korea), while humidity was managed using a humidifier (DE-9090UH, Zhongshan Xinhao Electrical., Ltd., Zhongshan, China) and a dehumidifier (EDHA11W3, WINIA). The photoperiod involved daylight conditions from 0900 to 1900 h.

### 4.3. Sampling and Analysis

#### 4.3.1. Feed and Water Intake and Milk Yield

Feed was provided twice daily (0800 h and 1400 h), and fresh water was offered five times daily to fulfill the nutritional needs of the lactating Holstein cows in each group. The DMI and water intake (WI) were measured using a scale machine (GL-6000S, G-Tech International, Seoul, South Korea) in a large trough using the remaining amount of feed and water. The milk yield was recorded daily during the experimental period.

#### 4.3.2. Milk Characteristics

We comprehensively analyzed 25 mL milk samples collected from both the morning and afternoon milkings. The samples were obtained using a mixed (5:5) methodology on the 3rd (adaptation), 7th (HS), and 14th day (HS with ACT supplementation) in each experimental group and were treated with milk preservatives (Broad Spectrum Microtabs II, D&F Control Systems Inc., Son Ramon, CA, USA). These composite milk samples were then subjected to storage at 4 °C to facilitate precise analysis of their constituent components. A MilkoScan FT1 instrument (FossAlle 1 DK-3400 Hilleroed, Hillerød, Denmark) was employed for this analytical process. In accordance with the NRC [93] guidelines, this paper reports the computation of 3.5%-fat-corrected milk (3.5% FCM) and energy-corrected milk (ECM) concentrations. The formula used to calculate the 3.5% FCM yield was [0.4324 × milk yield (kg/d)] + [16.216 × fat yield (kg/d)], whereas the ECM yield was calculated as [0.327 × milk yield (kg/d) + 12.95 × fat yield (kg/d) + 7.2 × protein yield (kg/d)].

#### 4.3.3. Blood Profiles

Blood samples were obtained from the jugular vein of the bovines at 1400 h on the 3rd (adaptation), 7th (HS), and 14th day (HS with ACT supplementation) in each experimental group. Blood samples were collected using ethylenediaminetetraacetic acid (EDTA)-treated vacutainers (Becton-Dickinson, Franklin Lakes, NJ, USA) and hematologically analyzed using a VetScan HM2 Hematology System (270405, Abaxis, Sac City, IA, USA). Nonheparinized vacutainers (BD Vacutainer, Plymouth, UK) and heparin tubes (Becton-Dickinson, Belliver Indsutril Estate, UK) were used for blood metabolite and hormone analysis, respectively. To analyze the metabolites and hormones, serum and plasma samples were separately obtained from nonheparine and heparin tubes after centrifugation at 2000× *g* for 15 min at 4 °C. Serum and plasma samples were separated, 500 µL each, into 1.5 mL Eppendorf tubes (Eppendorf AG, Hamburg, Germany) and then stored at −80 °C in a deep freezer (U9280-0002, Mississauga, ON, Canada). Serum was extracted for blood metabolite analysis using a Hitachi automatic chemical analyzer (Model 7180, Hitachi, Gyeonggi province, South Korea). For the analysis of cortisol, haptoglobin, and ghrelin levels, plasma samples were collected from heparinized blood tubes (Becton-Dickinson, Belliver Industril Estate, Plymouth, UK) and subjected to ELISA using specific kits (CSB-E13064B, CUSABIO, Houston, TX, USA; MBS739905, MyBioSource, San Diego, CA, USA; EB0066, FineTest, Wuhan, China). The cortisol, haptoglobin, and ghrelin levels in plasma were quantified at a wavelength of 450 nm using a spectrophotometer (PMT49984, BioTek Instruments Ltd., Winooksi, VT, USA). The inter-assay coefficients of variation for cortisol, haptoglobin, and ghrelin were 8%, 10%, and 4.25%, respectively, whereas the intra-assay coefficients of variation were 10%, 8%, and 5.12%, respectively.

#### 4.3.4. Peripheral Blood Mononuclear Cell (PBMC)

For the analysis, PBMCs were stored at room temperature in heparin tubes (Becton-Dickinson, Belliver Industril Estate, UK). We transferred 3 mL of Histopaque-1077 (RNBF8871, Sigma-Aldrich Co. LLC, St. Louis, MO, USA) and 3 mL of blood to a 15 mL conical tube. Subsequently, we centrifuged the mixture at 400× *g* for 30 min at 20 °C and extracted 2 mL of the white buffy coat layer. Only the buffy coat was placed in a 15 mL tube containing 8 mL of PBS (P2004, Biosesang, Yongin, South Korea), which was centrifuged at 300× *g* for 10 min at 20 °C. After the centrifugation, we removed the supernatant, leaving the pellet that had settled to the bottom, to which we added 1 mL of Trizol (TR 118, Molecular Research Center, Cincinnati, OH, USA) to release the pellet, which we transferred to a 2 mL tube (Eppendorf AG, Hamburg, Germany).

#### 4.3.5. Hair Follicle

Hair follicle samples were extracted from the tails of early lactating Holstein cows and stored in RNAlater^TM^ Solution (LT-02241, Invitrogen by thermos fisher scientific Baltics UAB. V.A. Graiciuno 8, Vilnius, Lithuania) at room temperature until analysis. The hair follicle was cut at its end to obtain at least 30 pieces, which were subsequently transferred into a 2 mL Eppendorf tube containing Trizol reagent. The samples were homogenized using an IKA T10 basic HOMOGENIZER WORKCENTER for one minute and subjected to the same RNA extraction and gene expression procedures [96].

#### 4.3.6. RNA Extraction from PBMC and Hair Follicle

Chloroform reagent (AH049-4, Honeywell) was added to each PBMC and follicle sample that had been mixed with Trizol reagent in a 2 mL Eppendorf tube. The solution underwent mechanical agitation through vortexing and was subsequently subjected to an incubation period of 2 min at room temperature in accordance with the experimental procedure. Following centrifugation at 13,000× *g* for 15 min at 4 °C, the resulting supernatant was carefully collected and transferred into a 1.5 mL tube. Isopropanol reagent (I9516, Sigma-Aldrich Co. LLC) was placed in the same tube, vortexed, and placed at room temperature for 10 min. The tube was then centrifuged again at 13,000× *g* for 10 min at 4 °C. After removing the supernatant only, to the white pellet that had settled to the bottom of the tube, we added 1 mL of 75% ethanol reagent (1.00983.1011, Sigma-Aldrich Co. LLC), which we then centrifuged at 7500× *g* for 10 min at 4 °C. After removing all the supernatant, the remaining liquid was removed with a vacuum dryer for 15 min, and 20 µL of diethyl pyrocarbonate water was added to each tube. A heating block was applied for 10 min. The RNA concentration was measured using a Nano drop 1000 Spectrophotometer (Thermo Scientific, Waltham, MA, USA) with an RNA-40 module; the samples were stored in a −80 °C freezer until the next analysis. For the cDNA synthesis and RT-PCR analysis, we followed the procedure reported by [97].

In this study, the expressions of HSP70 and HSP90 genes in PBMCs were quantified and normalized against those of three reference genes: glyceraldehyde-3-phosphate dehydrogenase (GAPDH), ribosomal protein s15a (RPS15A), and beta-2 microglobulin (B2M). This normalization was performed using the geometric mean of the relative quantities of these adjusted reference genes. Notably, in Indian cattle and buffaloes from tropical regions that are characterized by high HS, the stability of these reference genes in PBMCs is optimal [97]. Furthermore, for the qRT-PCR analysis of hair follicles, the HSP90 gene expression was normalized using GAPDH as the sole reference gene, applying the geometric mean of its corrected relative quantity [97]. The primers used in the qRT-PCR analyses are listed in Table 8. These primers were designed via the NCBI’s online system and sourced from Bioneer Co. (Bioneer, Seoul, South Korea).

#### 4.3.7. Physiological Indicators

Heart rate (HR, bpm) and rectal temperature (RT, °C) were recorded to study the changes that occur in the physiological parameters of cattle exposed to HS. HR and RT were measured on the 3rd (adaptation), 7th (HS), and 14th day (HS with ACT supplementation) in each experimental group at 0900 h and 1400 h, when temperature and humidity were high, when the conditions mimicked those in South Korea. Rectal temperature (RT) was measured using a TES 1300, K-Type thermometer (20593190, TES, Taipei, Taiwan) via insertion into the rectum for 1 min. The cow’s heart rate was assessed using a stethoscope for 1 min while in a stable condition.

### 4.4. Statistical Analysis

Milk yield and feed and water intake were analyzed using a repeated-measures analysis through the mixed procedure in SAS (Studio Version, SAS Institute Inc., Cary, NC, USA). In the model, the individual cow was considered the random effect. The factors included were HS, ACT, and their interaction (HS × ACT). Multiple comparisons within the model were conducted without adjustments, which is consistent with the approach followed by Rothman [98]. Each variable was evaluated for four covariance structures: autoregressive order 1, variance components, compound symmetry, and unstructured covariance. The covariance structure resulting in the lowest Akaike information criterion value was selected to compare mean values for each trait. Two-way analysis of variance was used to analyze milk characteristics, blood parameters, physiological indicators, and the HSP family gene expressions in PBMCs and hair follicles. The model included the treatment and animal as potential sources of variation. The significance of the variance between means was tested using Tukey’s test. Statistical significance was inferred when the *p*-value was less than 0.05, whereas a value between 0.05 and 0.10 suggested a tendency toward significant. The standard error of the mean is reported.

## 5. Conclusions

Lactating Holstein cows exposed to HS conditions, after being fed an ACT supplement (30 g/day), demonstrated improvements in various parameters, including DMI, WI, milk yield and protein, physiological indicators (HR and RT), blood hematology (PCT, EOS, MONO, and LUC), metabolites (albumin and glucose), hormone (cortisol and haptoglobin) and HSP70 gene expression in PBMCs.

These results indicate that ACT supplementation may serve as an effective antistress supplement for lactating Holstein cows exposed to HS, as it was found to improve their health, productivity, and the physiological stress response. However, further investigations are necessary to fully evaluate the effects of ACT supplementation on lactating Holstein cows under external environmental HS conditions.

## Figures and Tables

**Figure 1 ijms-25-01217-f001:**
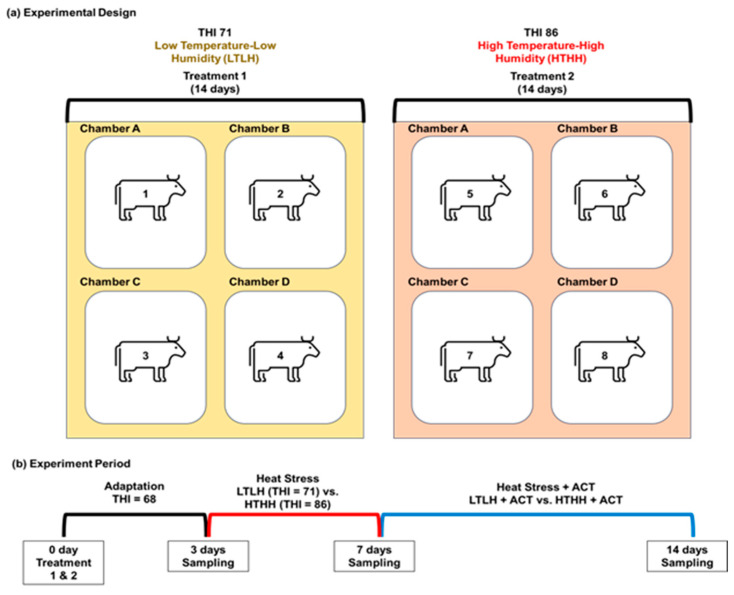
Experimental (**a**) design and (**b**) period. The schematic provides a detailed overview of the experimental framework conducted within climate chambers, focusing on heat stress (HS) treatments: LTLH and HTHH, supplemented with rumen-protected L-tryptophan (ACT). Eight early-lactation Holstein cows were systematically randomized into two balanced groups. In the initial adaptation phase, the subjects were subjected to a temperature–humidity index (THI) of 68 for a duration of three days. Following this adaptation phase, a designated four-day treatment regimen was applied. Thereafter, a seven-day exposure to the specified HS combined with ACT was initiated. After concluding the LTLH treatment phase, the HTHH treatment was implemented, utilizing a comparable methodology but integrating an alternative cohort of early lactating Holstein cows. THI: temperature–humidity index.

**Table 1 ijms-25-01217-t001:** Relationship between rumen-protected L-tryptophan and intake under heat stress (HS) conditions in early lactating Holstein cows.

	LTLH			HTHH			SEM	*p*-Value		
	Adaptation	HS	HS + ACT	Adaptation	HS	HS + ACT		HS	ACT	HS × ACT
DMI, kg/d	40.74	40.10	40.51	39.73	30.11	36.41	0.843	0.0087	0.0008	0.0018
WI, kg/d	89.87	99.67	99.38	76.40	113.76	80.59	2.110	<0.0001	<0.0001	<0.0001
F:W	0.45	0.40	0.42	0.52	0.28	0.45	0.013	0.0019	0.0025	0.0297
Milk yield, kg/d	39.63	40.63	41.15	36.06	30.89	31.15	0.773	0.0083	0.0262	0.0880

Values are expressed as means ± SEM (*n* = 4). LTLH, low-temperature, low-humidity, 25 °C, 35~50%; HTHH, high-temperature, high-humidity, 31 °C, 80~95%; HS, heat stress; ACT, rumen-protected L-tryptophan; SEM, standard error mean; DMI, dry matter intake; WI, water intake; F:W, ratio of feed and water.

**Table 2 ijms-25-01217-t002:** Relationship between rumen-protected L-tryptophan and milk characteristic under heat stress (HS) conditions in early lactating Holstein cows.

	LTLH			HTHH			SEM	*p*-Value		
	Adaptation	HS	HS + ACT	Adaptation	HS	HS + ACT		HS	ACT	HS × ACT
Milk protein, %	2.81	2.82	2.81	2.89	2.81	3.04	0.025	0.1227	0.0523	0.0379
Milk fat, %	4.46	4.13	4.65	4.66	3.68	3.72	0.317	0.2012	0.7293	0.7735
Lactose, %	5.05	4.98	4.99	4.98	4.96	4.90	0.027	0.5961	0.821	0.6476
SNF, %	8.34	8.41	8.29	8.42	8.44	8.48	0.033	0.3701	0.6384	0.3553
Milk protein, kg/d	1.02	1.16	1.12	1.04	0.90	1.08	0.039	0.3158	0.5113	0.2742
Milk fat, kg/d	1.68	1.66	1.92	1.99	1.12	1.30	0.116	0.0418	0.4235	0.8834
Lactose, kg/d	1.89	2.03	2.06	1.80	1.53	1.75	0.081	0.1433	0.5315	0.6347
SNF, kg/d	3.11	3.42	3.40	3.05	2.60	3.02	0.127	0.1780	0.5292	0.4954
3.5% FCM	43.37	44.46	48.90	47.81	31.49	36.50	2.264	0.0306	0.3684	0.9561
ECM	41.27	43.12	46.34	45.03	31.09	36.24	1.977	0.0368	0.3653	0.8335
SC, 10^3^/mL	90.00	75.50	47.00	72.75	39.00	95.00	20.228	0.4930	0.2688	0.9215
MUN, mg/dL	15.96	15.45	13.15	13.38	13.73	16.33	0.369	0.7089	0.8681	0.0136
Acetone, mM	0.03	0.04	0.05	0.10	0.05	0.00	0.012	0.6792	0.5572	0.3675
BHB, mM	0.06	0.08	0.08	0.08	0.07	0.05	0.005	0.3501	0.4846	0.4846
Cas.B	2.08	2.18	2.07	2.21	2.24	2.30	0.028	0.0936	0.7647	0.2222

Values are expressed as means ± SEM (*n* = 4). LTLH, low-temperature, low-humidity, 25 °C, 35~50%; HTHH, high-temperature, high-humidity, 31 °C, 80~95%; HS, heat stress; ACT, rumen-protected L-tryptophan; SEM, standard error mean; SNF, solid—not fat; FCM, fat-corrected milk; ECM, energy-corrected milk; SC, somatic cell; MUN, milk urea nitrogen; BHB, beta-hydroxybutyrate; Cas.B, beta-casein.

**Table 3 ijms-25-01217-t003:** Relationship between rumen-protected L-tryptophan and blood hematology under heat stress (HS) conditions in early lactating Holstein cows.

	LTLH			HTHH			SEM	*p*-Value		
	Adaptation	HS	HS + ACT	Adaptation	HS	HS + ACT		HS	ACT	HS × ACT
WBC, 10^3^ cells/μL	8.46	7.73	7.98	9.85	8.40	10.30	0.417	0.2162	0.3022	0.4286
RBC, 10^6^ cells/μL	5.52	4.95	5.34	6.03	5.96	6.49	0.147	0.0039	0.1296	0.8111
HGB, g/dL	9.90	9.48	9.55	10.25	10.25	10.75	0.191	0.1112	0.5345	0.6454
HCT, %	26.53	24.00	25.75	27.40	26.83	29.10	0.619	0.0785	0.1755	0.8563
MCV, fL	48.08	48.65	48.18	45.50	45.05	44.93	0.554	0.0453	0.8153	0.8915
MCH, pg	17.90	19.53	17.88	17.03	17.23	16.65	0.351	0.0696	0.1805	0.5098
MCHC, g/dL	37.33	40.10	37.05	37.45	38.23	37.08	0.505	0.2846	0.1017	0.4461
PLT, 10^3^ cells/μL	666.00	606.00	549.25	637.75	659.50	524.50	29.396	0.9484	0.2070	0.6001
MPV, fL	6.83	9.00	6.43	5.90	6.13	6.38	0.376	0.1081	0.1849	0.1110
PDW, %	58.33	71.55	60.53	51.48	51.13	57.68	3.077	0.2282	0.7668	0.2520
PCT, %	0.45	0.56	0.34	0.37	0.40	0.33	0.031	0.3418	0.0600	0.3299
NEUT, %	39.88	33.68	37.48	40.48	39.00	55.48	2.839	0.2053	0.1418	0.3500
LYM, %	47.63	55.20	47.30	48.40	48.50	35.00	2.798	0.3367	0.1292	0.6828
MONO, %	6.45	4.40	7.60	4.88	5.13	4.83	0.494	0.5352	0.2417	0.1611
EOS, %	4.63	5.45	5.15	4.50	5.95	3.23	0.360	0.3027	0.0880	0.1657
LUC, %	0.58	0.23	1.25	0.90	0.48	0.68	0.133	0.4240	0.0656	0.2040
BASO, %	0.88	1.05	1.20	0.88	0.93	0.78	0.052	0.0594	1.0000	0.2109
NEUT, 10^3^ cells/μL	3.29	2.53	3.00	4.11	3.27	6.07	0.428	0.1271	0.1050	0.2407
LYM, 10^3^ cells/μL	4.14	4.35	3.81	4.67	4.09	3.30	0.273	0.8313	0.3536	0.8656
MONO, 10^3^ cells/μL	0.51	0.34	0.56	0.47	0.42	0.48	0.027	0.2152	0.0317	0.2009
EOS, 10^3^ cells/μL	0.41	0.41	0.42	0.44	0.52	0.29	0.035	0.8839	0.2146	0.2050
LUC, 10^3^ cells/μL	0.05	0.02	0.10	0.08	0.03	0.08	0.011	0.3207	0.0325	0.4886
BASO, 10^3^ cells/μL	0.08	0.08	0.10	0.09	0.08	0.08	0.006	0.7909	0.7342	0.4990

Values are expressed as means ± SEM (*n* = 4). LTLH, low-temperature, low-humidity, 25 °C, 35~50%; HTHH, high-temperature, high-humidity, 31 °C, 80~95%; HS, heat stress; ACT, rumen-protected L-tryptophan; SEM, standard error mean; WBC, white blood cell; RBC, red blood cell; HGB, hemoglobin; HCT, hematocrit; MCV, mean corpuscular volume; MCH; mean corpuscular hemoglobin; MCHC, mean corpuscular hemoglobin concentration; RDW, red blood cell distribution width; HDW, hemoglobin distribution width; PLT, platelet; MPV, mean platelet volume; PDW, platelet distribution width; PCT, plateletcrit; NEUT, neutrophil; LYM, lymphocyte; MONO, monocyte; EOS, eosinophil; LUC, large unstained cell; BASO, basophil.

**Table 4 ijms-25-01217-t004:** Relationship between rumen-protected L-tryptophan and blood metabolites and hormones under heat stress (HS) conditions in early lactating Holstein cows.

	LTLH			HTHH			SEM	*p*-Value		
	Adaptation	HS	HS + ACT	Adaptation	HS	HS + ACT		HS	ACT	HS × ACT
Albumin, g/dL	3.25	3.12	3.12	3.26	3.47	3.16	0.029	0.0001	0.0049	0.0067
BUN, mg/dL	19.00	18.75	15.25	14.25	16.25	20.25	0.758	0.7054	0.8916	0.0521
CA, mg/dL	9.30	9.13	9.13	9.20	9.38	9.70	0.107	0.3252	0.5502	0.5502
CHO, mg/dL	230.25	227.75	241.50	217.50	239.50	221.75	11.003	0.9377	0.9463	0.5974
Globulin, g/dL	4.00	3.78	3.78	3.50	3.82	3.47	0.083	0.8103	0.4248	0.4248
Glucose, mg/dL	64.00	58.75	56.50	62.75	68.75	67.00	0.964	0.0001	0.0307	0.7728
GOT, U/L	73.25	61.50	57.75	76.25	67.75	76.50	3.547	0.2058	0.7763	0.4799
r-GT, U/L	31.00	29.00	27.50	26.75	27.25	24.75	1.716	0.8737	0.6650	0.9136
MG, mg/dL	2.30	2.53	2.45	2.20	2.30	2.30	0.046	0.1045	0.7366	0.7366
NEFA, uEq/L	311.00	263.50	207.75	175.50	231.75	323.00	24.379	0.9243	0.4593	0.9283
IP, mg/dL	5.40	6.13	6.45	5.68	5.25	5.78	0.181	0.2172	0.3394	0.8201
Total protein, g/dL	7.25	6.90	6.89	6.76	7.29	6.63	0.102	0.9089	0.2010	0.2075
Cortisol, ng/mL	80.70	107.05	89.42	95.97	192.22	138.08	8.641	0.0001	0.0012	0.0687
Haptoglobin, ng/mL	396.45	437.81	408.94	401.57	430.97	389.80	1.570	0.1586	0.0765	0.7460
Ghrelin, pg/mL	126.63	118.18	116.31	128.93	106.62	108.81	2.012	0.0001	0.8116	0.0094

Values are expressed as means ± SEM (*n* = 4). LTLH, low-temperature, low-humidity, 25 °C, 35~50%; HTHH, high-temperature, high-humidity, 31 °C, 80~95%; HS, heat stress; ACT, rumen-protected L-tryptophan; SEM, standard error mean; BUN, blood urea nitrogen; CA, calcium; CHO, cholesterol; GOT, glutamic-oxaloacetic transaminase; r-GT, gamma-glutamyltranspeptidase; MG, magnesium; NEFA, non-esterified fatty acid; IP, inorganic phosphorous.

**Table 5 ijms-25-01217-t005:** Relationship between rumen-protected L-tryptophan and heat shock protein gene expression under heat stress (HS) conditions in early lactating Holstein cows.

	LTLH			HTHH			SEM	*p*-Value		
	Adaptation	HS	HS + ACT	Adaptation	HS	HS + ACT		HS	ACT	HS × ACT
**PBMC**										
HSP70	0.50	0.60	0.68	0.65	0.52	0.86	0.028	0.5812	0.0011	0.0238
HSP90	0.74	0.90	0.98	0.86	0.73	0.83	0.032	0.1854	0.2828	0.9641
**Hair follicle**										
HSP90	1.82	1.43	0.90	1.08	1.84	1.22	0.180	0.7165	0.2687	0.9305

Values are expressed as means ± SEM (*n* = 4). LTLH, low-temperature, low-humidity, 25 °C, 35~50%; HTHH, high-temperature, high-humidity, 31 °C, 80~95%; HS, heat stress; ACT, rumen-protected L-tryptophan; SEM, standard error mean; PBMC, peripheral blood mononuclear cell; HSP, heat shock protein.

**Table 6 ijms-25-01217-t006:** Relationship between rumen-protected L-tryptophan and physiological indicators under heat stress (HS) conditions in early lactating Holstein cows.

	LTLH			HTHH			SEM	*p*-Value		
	Adaptation	HS	HS + ACT	Adaptation	HS	HS + ACT		HS	ACT	HS × ACT
Heart rate, beat/min	67.00	79.00	77.00	67.25	89.25	85.00	1.826	0.0001	0.0242	0.7454
Rectal temperature, °C	38.18	38.33	38.13	38.23	39.18	38.73	0.094	<0.0001	<0.0001	0.6740

Values are expressed as means ± SEM (*n* = 4). LTLH, low-temperature, low-humidity, 25 °C, 35~50%; HTHH, high-temperature, high-humidity, 31 °C, 80~95%; HS, heat stress; ACT, rumen-protected L-tryptophan; SEM, standard error mean.

**Table 7 ijms-25-01217-t007:** Examination of the chemical compositions and amino acid profiles of the experimental feed.

Items	TMR	Concentrates	ACT
Composition, % (DM basis)			
Dry matter	61.91	88.84	
Crude Protein	6.06	18.06	
Crude Fat	1.11	2.75	
Crude Fiber	8.06	6.44	
Crude Ash	3.15	6.17	
Calcium	0.37	0.61	
Phosphorus	0.17	0.50	
NDF	21.53	21.61	
ADF	11.30	10.71	
NDFn	19.36	18.79	
NDIP	2.17	2.82	
ADIP	0.52	0.73	
tdNFC	71.67	55.27	
tdCP	5.46	17.77	
tdFA	0.11	1.75	
Amino acids, % (DM basis)			
Tryptophan	0.06	0.19	0.16
Threonine	0.22	0.60	
Serine	0.26	0.77	
Proline	0.33	0.94	
Valine	0.27	0.70	
Isoleucine	0.17	0.51	
Leucine	0.37	1.26	
Tyrosine	0.10	0.39	
Methionine	0.06	0.18	
Cystine	0.13	0.41	
Lysine	0.25	0.52	
Glycine	0.24	0.71	
Alanine	0.30	0.84	
Arginine	0.33	1.01	
Glutamic acid	0.84	2.83	
Aspartic acid	0.49	1.34	
Histidine	0.10	0.33	
Phenylalanine	0.22	0.68	

TMR, total mixed ratio; DM, dry matter; NDF, neutral detergent fiber; ADF, acid detergent fiber; NFC, neutral fiber; NDF, neutral detergent fiber; NDIP, neutral detergent insoluble crude protein; ADIP, acid detergent insoluble crude protein; tdNFC, truly digestible NFC; tdCP, truly digestible crude protein; tdFA, truly digestible fatty acid.

**Table 8 ijms-25-01217-t008:** Primer sequences design in bovine.

Gene	Accession Number ^1^	Sequence (5′ to 3′)	Length (bp)
HSP70	U09861	F: TACGTGGCCTTCACCGATAC R: GTCGTTGATGACGCGGAAAG	171
HSP90	NM_001012670	F: GGAGGATCACTTGGCTGTCA R: GGGATTAGCTCCTCGCAGTT	177
RPS15A	NM_001037443.2	F: CCGTGCTCCAAAGTCATCGT R: GGGAGCAGGTTATTCTGCCA	200
B2M	NM_173893.3	F: GACACCCACCAGAAGATGGA R: CAGGTCTGACTGCTCCGATT	125
GAPDH	NM_001034034.2	F: GGCAAGGTCATCCCTGAG R: GCAGGTCAGATCCACAACAG	166

HSP70, heat shock protein 70; HSP90, heat shock protein 90; RPS15A, ribosomal protein S15a; B2M, beta-2-macroglobulin; GAPDH, glyceraldehyde 3-phosphate dehydrogenase. ^1^ Database protein names and accession number: NCBI (http://www.ncbi.nih/gov (accessed on 5 December 2019)).

## Data Availability

The data were not deposited in an official repository. The data that support the study findings are available from the authors upon request.

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
