# Peer review of "Effects of Rumen-Protected L-Tryptophan Supplementation on Productivity, Physiological Indicators, Blood Profiles, and Heat Shock Protein Gene Expression in Lactating Holstein Cows under Heat Stress Conditions"

_ijms, 2024, doi:10.3390/ijms25021217_

Round 1

Reviewer 1 Report

Comments and Suggestions for Authors

Manuscript ID: ijms-2805958

Title: Effects of Rumen-Protected L-tryptophan Supplementation on Productivity, Physiological, Blood Profiles, and Heat Shock Protein Gene Expression in Lactating Holstein Cows Under Heat Stress Conditions

The manuscript needs some revisions, because there are some aspects of the work that should be corrected and improved. Please, review the following recommendations:

In title: Change "physiological" to "physiological indicators". " physiological" alone is not correct

English grammar and style must be corrected throughout the article.

Improve the introduction section, providing the problem.

The study was well conducted but suffers from a small sample size. I am concerned that is a major limitation of this study, and as such is less informative.

In the discussion section, please be more specific, discuss your study with other similar studies, and state the superiorities of your research compared to previous ones. Please enrich the discussion by addressing the study's limitations and practical applications.

In abstract: Write the whole name for these abbreviations " PCT ", " EOS", " MONO", " LUC"

Line 21: Change "40 ± 9 day" to "40 ± 9 days"

Line 50: Add "the" before "leaky gut"

Line 76: Change "production performance " to " productive performance "

The authors mentioned the purpose of the study in more than one sentence: Line 70 (we aimed), line 72 (The aim of this investigation), line 77 (The purpose of this study). So please write the aim of study in one paragraph

Lines 85, 88, 89, 105, 155: Add "was" before "decreased"

Lines 86, 91: Add "was" before "increased"

In Table 1: " WI, kg/d" check units "kg or L"

Line 132: Change "amounts of monocytes" to "numbers of monocytes "

Lines 154: Change "Blood Metabolite and Hormone" to "Blood Metabolites and Hormones"

Line 155: Add "were" before "increased"

Line 157: Add "were" before "decreased"

Lines 166: Change "blood metabolite and hormone" to "blood metabolites and hormones"

In Table 5: why authors determined HSP70 in PBMC only and not in Hair follicle

Line 211: Change "environmental temperatures" to " environmental temperature"

Line 227: Change "growth, gut integrity" to " growth and gut integrity"

Lines 255-257: Rewrite this sentence

Lines 287-288: Rewrite this sentence

Line 291: " now know"?

Line 411: Change " correlating to" to " correlating with"

Line 470: Change " HSP70 genes" to " HSP70 gene"

In Figure 1: Change " Experiment Design" to " Experimental Design"

In line 534: Change " Experiment Design" to " Experimental Design"

Lines 585, 598 and 679: Use Superscript  "3rd (adaptation), 7th (HS), and 14th day"

Comments on the Quality of English Language

Moderate editing of English language required

Author Response

Once again we are thankful to the reviewer for the insights comments that all were taken into consideration to improve the quality of our presentation.

Reviewer 2 Report

Comments and Suggestions for Authors

The study has strong scientific significance. The manuscript was impeccably written. I have only few corrections to suggest.

L19: L-tryptophan

L30: define PCT

 Table 1-5:  It may be clearer to label this ‘HS’ sub-heading as Heat stress status -Normal and Heat stress status-stressed. That way, readers are not confused.

L133: ad determined.

L135: eosinophil percentage (EOS)

Write Heat stress in table headings.

L480: increased in heat load after meal

L495: Consistent with the findings of prior research, our findings demonstrate 495 that L-tryptophan can considerably decrease the RT in chicken when supplemented at 496 doses ranging from 0~50 mg/kg BW, as reported by Badakhshan et al., [92].

Authors did not use chicken for the current study. This sentences need correction.

 Lastly, the methodology should have actually run the two low and high stress treatments at the same time and then repeat rather than run the low heat then the high heat protocol at different times. 

Comments on the Quality of English Language

The language expression is very clear and lucid most times. Only few corrections are suggested. 

Author Response

(The authors gave the same response as above.)

Reviewer 3 Report

Comments and Suggestions for Authors

The manuscript fits into the scope of the journal and the special issue.  Generally, it  provides interеsting data about the L-Trp when protected on some physiological parameters in dairy cows under heat stress. The study has well defined aim. The introduction is very well structured and points towards the need of such study and how it is beyond the state of the art. The cited references are relevant.

The material and methods are described in sufficient details. However my main concern is about the entire experimental design. The study contains only 4 cows in a group which is below the minimum for statistical evaluation. Moreover, the study also includes genetic analysis and 4 animals per groups are quite few for such kind of work. In my opinion based on such a small sample size, it is not possible to draw sound conclusions. May be it should be transformed as preliminary study?

On the other hand, the results are well described. I have also a remark about the tables. Below the Tables the authors withe that the results are presented as Mean plus minus SEM. This is not true. They present SEM, but this is actually the Root Mean Square Error for the model.

The discussion is also quite well done. The references are relevant and timely.

Author Response

(The authors gave the same response as above.)

Round 2

Reviewer 1 Report

Comments and Suggestions for Authors

The revised paper merits the final acceptance

Reviewer 3 Report

Comments and Suggestions for Authors

The authors have taken into account the remarks and have provided thorough explanations.